# Transportation Mode Detection Using Temporal Convolutional Networks Based on Sensors Integrated into Smartphones

**DOI:** 10.3390/s22176712

**Published:** 2022-09-05

**Authors:** Pu Wang, Yongguo Jiang

**Affiliations:** College of Computer Science and Technology, Ocean University of China, Qingdao 266100, China

**Keywords:** deep learning, temporal convolutional networks, activity recognition, transportation mode detection

## Abstract

In recent years, with the development of science and technology, people have more and more choices for daily travel. However, assisting with various mobile intelligent services by transportation mode detection has become more urgent for the refinement of human activity identification. Although much work has been done on transportation mode detection, accurate and reliable transportation mode detection remains challenging. In this paper, we propose a novel transportation mode detection algorithm, namely T2Trans, based on a temporal convolutional network (i.e., TCN), which employs multiple lightweight sensors integrated into a phone. The feature representation learning of multiple preprocessed sensor data using temporal convolutional networks can improve transportation mode detection accuracy and enhance learning efficiency. Extensive experimental results demonstrated that our algorithm attains a macro F1-score of 86.42% on the real-world SHL dataset and 88.37% on the HTC dataset, with an average accuracy of 86.37% on the SHL dataset and 89.13% on the HTC dataset. Our model can better identify eight transportation modes, including stationary, walking, running, cycling, car, bus, subway, and train, with better transportation mode detection accuracy, and outperform other benchmark algorithms.

## 1. Introduction

In recent years, transportation mode detection has many new applications in all aspects of daily human life [1,2,3]. For example, identification of detailed human environment in the field, carbon footprint calculation [4], and mining population flow patterns [5]. Smartphones have been inseparable from people, and many mobile phone models have now been equipped with various sensors, which can provide rich sensor data.

The IMU data can be accessed passively via the operating system or app and can be applied to research on reducing carbon dioxide production, uncovering population movement patterns, designing energy-efficient transport systems, building transport networks, traffic scheduling, predicting future traffic conditions, etc. In terms of carbon footprint calculations, as the sensor sits on a smartphone, it can collect the transportation mode of a large number of people over a long period of time. Transportation mode detection can motivate people to assess and reduce their carbon dioxide consumption [4]. In addition, some research centers organize volunteers to collect transportation mode detection samples, whose transportation modes are recorded almost every second by the sensors on the smartphone, for studies such as mining population movement patterns [5]. An energy-efficient transport system contributes to the country’s economic development and green environment. Transportation mode detection plays a key role in the design of energy-efficient transport systems. Researchers use sensors on experimenters’ mobile phones to collect data to build tools for traffic modeling, such as traffic networks and parking facilities. In addition, traffic congestion can be time and economically costly. Understanding human choices of traffic modes in large transport networks is crucial for urban congestion prediction and traffic scheduling [3]. In several studies on transportation mode detection, researchers have collected a large amount of IMU data and constructed intelligent systems. Given an arbitrary time and location, and combined with identified human transportation modes, the system can automatically simulate or predict future traffic conditions.

Many previous studies have simplified the modes of transportation into four categories: stationary, walking, bicycling, and vehicle-riding [6]. Unlike previous studies, to obtain more practical information, we used fine-grained modes of transportation: stationary, walking, running, cycling, car, bus, subway, and train.

In this paper, we propose a more accurate and concise data training model, T2Trans, based on a TCN for transportation mode detection. T2Trans is evaluated on the SHL dataset [7] and HTC dataset [8]. It should be noted that this study on transportation mode detection removed the limitations of previous GPS or motion sensors [5,9] and used IMU data that were more targeted at daily practicability. Methods also changed from old to new, from traditional machine learning based on manual feature extraction [9,10,11,12,13,14] to deep learning automatic feature extraction.

In the research of transportation mode detection, data pertinence and effectiveness are essential aspects of identifying people’s daily transportation mode detection. What kind of data to choose for mining is critical. Based on the GPS sensor on users’ mobile devices, L. Stenneth and O. Wolfson et al. used the knowledge of the underlying traffic networks to infer users’ transportation modes. A Roy et al. [15] investigated the use of GPS data to assess the role of geography in transportation mode detection. Integrating the geographic context in which mobility occurs can provide contextual clues that make models for predicting transportation mode detection more general. Chandrasiri G. et al. [16] investigated the application of GPS-based travel mode detection methods in energy-efficient transportation. Rule-based algorithms and heuristic-based approaches are the two main methods used to detect the four travel modes (walking, bus, motor vehicle, and motorcycle). However, the high power consumption of GPS and the limitation of data classification limit more detailed classification and recognition research. Nowadays, with the rapid development and broad application of smartphones, a variety of sensors on mobile phones can collect more diverse data, which provides a more promising basis for us to realize transportation mode detection. Recently, some researchers have introduced deep learning technology into transportation mode detection, such as recursive neural networks [17], convolutional neural networks [18], and long and short memory recursive neural networks [19]. However, existing studies are only evaluated on small datasets, with fewer transportation modes and accuracy that cannot meet application requirements. Model optimization of IMU data still faces significant challenges:Availability: The technology used in current transportation mode detection is generally based on GPS, but the common problem is excessive energy consumption and poor adaptability and stability. Secondly, in the process of motor vehicle driving, due to the uncertainty of road conditions and the differences in drivers’ driving habits, etc., it will also lead to the difficulty of getting the desired effect on the distinction between motor vehicles.Lightweight: IMU data sources have the advantage of being lightweight. Now many models have too many parameters, and we need the model parameters to be less, faster running technology.Expert knowledge: Transportation mode detection does not require manual extraction of features and does not rely on manual challenges.

Therefore, it is necessary to design a better transportation mode detection model to alleviate the impact of poor model design and identify eight more detailed modes of transportation: stationary, walking, running, cycling, car, bus, subway, and train. 

Several recent studies have used deep learning models to capture complex nonlinear dependencies [19]. However, they are still limited in terms of the model’s accuracy and the model design’s complexity. In our study, we designed T2Trans based on the TCN algorithm to investigate the problem of transportation mode detection. To address these challenges, we have made the following contributions:We identified eight specific modes of transportation: stationary, walking, running, cycling, car, bus, subway, and train. We exploit TCNs for transportation feature representation learning. Through TCNs and a parallel implementation of convolution, we alleviate the computation burden. TCNs change the receptive field by increasing the number of layers and changing the expansion coefficient and filter size. The gradient of the TCN is in the direction of network depth. For long sequences, a TCN connected by residuals is more stable.We use two IMU datasets, which are lightweight, and our proposed T2Trans can comprehensively identify features that differ between traffic modes by analyzing sensing data at a fine-grained level. Moreover, the T2Trans model has fewer parameters and runs faster.We evaluate our new approach on two large public datasets, the SHL dataset [7] and the HTC dataset [8]. Experimental results show that T2Trans is significantly better than baseline algorithms, including DT, RF, XGBOOST, CNN, MLP, MLP + LR, Bi-LSTM, CNN + LSTM, and other transportation mode detection methods. Reasonable scalability of T2Trans has also been demonstrated. Extensive results showed that T2Trans achieved the best performance on F1-scores compared to all baselines, with an improvement of 5.94% over the best baseline on the SHL dataset and a 6.40% improvement over the best baseline on the HTC dataset.

## 2. Related Work

In recent years, several researchers used GPS data for transportation mode detection. L. Stenneth et al. [16], based on GPS sensors on users’ mobile devices, inferred users’ modes of transportation with the knowledge of underlying transportation networks and inferred various modes of transportation with an accuracy rate of more than 93.5%, including car, bus, ground train, walking, bicycle and fixed mode. S. Dabiri and K. Halip et al. [18] used a convolution neural network to infer the transportation mode from the GPS track, in which the mode is marked as walking, cycling, bus, driving, and train, achieving the highest accuracy of 84.8%. H. Liu et al. [19] combined the feature learning of timestamp, latitude, and longitude obtained from GPS sensors and proposed a classification framework of track transportation modes based on an end-to-end bidirectional LSTM classifier. Although GPS data can be used from the perspective of facilitating transportation mode detection [9], the high power consumption and environmental limitations of GPS affect the progress of many pieces of research in this aspect. 

GPS can suffer from high power consumption, and if the user is in an environment such as a subway where the connection between GPS and satellites is blocked, data acquisition can be difficult and inaccurate. Therefore, some existing work has used other environment-dependent sensors for transportation mode detection. A. Jahangiri et al. [13] utilized smartphone sensors such as accelerometers, gyroscopes, and rotation vector sensors to identify modes of transportation, including driving a car, riding a bicycle, taking a bus, walking, and running. I. Drosouli et al. [20] used machine learning techniques to perform transportation mode detection on cell phone sensor data. The classifier was developed without dimensionality reduction and then used the PCA algorithm to use a more lightly modeled model. After dimensionality reduction, the best performing algorithm achieved very good classification results, while training time was significantly reduced. S. Hemminki et al. [12] adopted supervised learning methods in machine learning to develop a multi-class classifier and evaluated the performance in different ways. RF and SVM methods are found to produce the best performance for identifying modes of transportation, including driving a car, cycling, riding a bus, walking, and running. Ashqar et al. [9] used an accelerometer sensor to estimate the gravity component measured by the accelerometer to improve the generalization and robustness of detection. In general, manual extraction of features from raw sensor data may be a heavy burden on users in most machine learning, and the accuracy and robustness of transportation mode detection remain improved.

With the continuous development and broad application of deep learning in recent years, compared with traditional machine learning, the upper limit of deep learning in transportation mode detection is higher. S. Dabiri et al. [18] predicted transportation modes based on original GPS trajectories and used a CNN architecture to mark walking, bicycle, bus, driving, and train modes. A four-channel input volume is created, including velocity, acceleration, jerkiness, and load-bearing rate. The method of early stop is adopted to select the best epoch number. Integrating the best CNN configuration achieves the highest accuracy of 84.8%. U. Majeed et al. [21] use multiple smartphone sensors for vanilla split learning of transportation mode detection. The researchers performed vanilla segmentation learning for transport pattern detection on a smartphone sensor-based dataset, demonstrating that the split neural network (SplitNN) performs similarly to a baseline typical deep neural network. C. Wang et al. [22] use a multimodal sensor integrated into a smartphone, combined with residuals and an LSTM recurrent network, for transportation mode detection. The researchers propose a transportation mode detection algorithm based on residuals and LSTM recurrent networks, using data collected by multiple lightweight sensors in cell phones. A residual unit is introduced to speed up training, and an attention model is used to learn different features and time steps to improve recognition accuracy. M. Ahmed et al. [23] established a convolutional neural network to determine the transportation mode using the acceleration sensor on a smartphone, achieving an accuracy as high as 94.48%. Jahangiri et al. [13] proposed a recurrent neural network (i.e., RNN) to classify eight modes of motion and transportation activities. The network is assigned to select the best universal classifier (random forest, decision tree, gradient lifting, etc.) to classify the active tags. It utilizes the rotation of acceleration and magnetometer values from mobile phone coordinates to earth coordinates to propose the characteristics of JERK and position insensitivity. L. Wang et al. [24] identified eight modes of transportation (stationary, walking, running, bicycle, bus, car, train, and subway) through the inertial sensor of smartphones, provided different types of input to the classifier, and adopted the post-processing scheme to improve the recognition performance. The convolutional neural network running on the raw data in the frequency domain achieves the best performance among all classifiers. H. Liu et al. [19] proposed a classification framework of trajectory transport mode based on an end-to-end bidirectional LSTM classifier, which automatically learns features from the trajectory and leverages them for classification. Meanwhile, a method to identify and adjust the outliers naturally contained in the trajectory data is applied in flexible route discovery based on GPS trajectory data [25]. Chen L. et al. and the Ito et al. [26] used FFT from acceleration and gyro sensor data to convert 5-s sensor sequence data into two-dimensional spectrographic images and applied the transfer learning method to pre-training models from other sensors labeled as hips and torsos. Ganbare A.M. et al. [27] used long- and short-term memory networks with common preprocessing steps, adopted two-layer LSTM to identify transportation modes, and achieved 63.68% accuracy with the internal test data set. A convolutional neural network (CNN) is used to learn appropriate and robust feature representation and further learn time-dependent features of feature vectors outputted by CNN through the LSTM network [28]. The CL-Transmode identification algorithm can accurately distinguish eight modes of transportation: walking, running, cycling, driving, bus, subway, train, or standing still. The accuracy of the CL-Transmode algorithm can reach 98.1% on the SHL data set containing barometric data. The authors of [29] proposed attention-based bidirectional long- and short-term memory (ABLSTM) for passive human activity recognition using WiFi CSI signals. The attention mechanism is exploited to assign different weights to all learned features. Practical experiments have been conducted to evaluate the performance of the proposed ABLSTM in human activity recognition, distinguishing between various activities, including lying down, falling, walking, running, sitting, and standing up, with an average recognition rate of 97%. Although it is energy saving compared with GPS, uneven WiFi access point density and GSM cellular size will significantly impact the experiment and require more stringent environmental requirements.

Unlike previous studies, we propose a TCN-based T2Trans, which uses multiple light sensors integrated into the mobile phone for transportation mode detection. TCN realizes convolution in parallel, making the network faster. Compared with other studies, we use TCN to change the receptive field. Especially for long sequences, less memory is occupied during training, and our algorithm attains a macro F1-score of 86.42% on the real-world SHL dataset and 88.37% on the HTC dataset.

## 3. Algorithm

### 3.1. Overview

In the data preprocessing step, all datasets are transformed into a unified matrix and fed into the input layer structure. In the model training step, preprocessed sensor data from the selected dataset is used as training data and is fed into the T2Trans model to optimize the trainable parameters in the T2Trans model. We consider the identification task of barometric pressure and three other inertial sensors: linear acceleration, gyroscope, and magnetometer. Each sensor contains three elements including the x, y, and z axes of the device, which were fused due to the unknown posture and orientation of the smartphone. To make the extracted temporal features have causal characteristics and achieve multi-scale feature extraction and receptive field enlargement, the convolution operation of the time convolution network [30] is carried out on a single sensor immediately after pretreatment, and then the feature representations from sensors are combined into one tensor. To directly transmit the underlying features across layers and enhance the gradient propagation capability, the convolution operation of the time convolution network is applied to the merged feature representation again. At last, representations with time dependencies are passed through multiple fully connected layers and finally produce the transportation mode estimation using Softmax. In the model inference step, testing data are leveraged to evaluate the performance of our proposed T2Trans model based on accuracy, precision, recall, and F1-score metrics.

### 3.2. T2Trans Model

In this section, we will introduce the overall architecture and details of the T2Trans model. The overall architecture of T2Trans is shown in Figure 1.
Multimodal Input Layer. All preprocessed sensor data are fed into the model from the multimodal input layer, defined as tensors $A_{n,\;d,k}$
where n denotes the total number of samples, k denotes the total number of units of all sensors, and d denotes the length of the selected sliding window (i.e., 500 samples correspond to a sampling period of 5 s, with a sampling frequency of 100 Hz). In this paper, k=10 represents linear acceleration axis X, Y, and Z; gyroscope axis X, Y, and Z; magnetometer axis-X, Y, and Z, and barometric pressure. $A_{n,\;d,10}$ is converted into 10 tensors denoted by $A_{n,\;d,1}$, which are fed into ten channel TCN layers, respectively. Barometric pressure is directly sent to the CTN layer, and the three elements of the other three sensors (i.e., linear acceleration, gyroscope, and magnetometer) are fused first and then fed into the channel TCN layer, as illustrated in Figure 2.Channel TCN Layer. A temporal convolutional network (TCN) is a variant of the convolutional neural network. A temporal convolutional network is a convolutional neural network model based on a traditional one-dimensional convolutional neural network and combined with causal convolution, extended convolution, and residual link. Preliminary empirical evaluations of TCN show that simple convolutional architectures exhibit better performance over a variety of tasks and datasets than conventional recursive networks such as LSTM [31] while demonstrating longer effective memory. TCN has flexible receptive fields and stable gradients and can map timing data to output sequences of the same length [32]. TCN uses the one-dimensional convolution kernel to sweep into the current time node and the historical time series data before the node for data processing along the network layer. We use an input sequence $x_{0}$, …, $x_{T}$, to predict some related outputs $y_{0}$, …, $y_{T}$ at each time period. In order to predict the output $y_{T}$ at some time t, we can only use those inputs that were previously observed: $x_{0}$, …, $x_{T}$. A sequence modeling network is any function $f:X^{T+1}\to Y^{T+1}$ that produces the mapping:

(1)$${\hat{y}}_{0},\ldots ,{\hat{y}}_{T}=f\left(x_{0},\ldots ,x_{T}\right)$$


Like RNN, the architecture of TCN can take a sequence of any length and map it to an output sequence of the same length. In addition, we emphasize how to use a combination of very deep networks (with the enhancement of residual networks) and extended convolution to build very long effective history sizes. We are now in the position to elaborate on the technical details of TCN in three ways: TCN uses causal convolution. The output at this moment is only convolved with the corresponding input at the moment in the previous layer and the input at the earlier moment and has nothing to do with the future moment. As shown in Figure 3a, the size of each convolution kernel denoted by k
is 3. TCN adds an extra length of zero padding to keep subsequent layers the same length as previous layers.TCN uses extended convolution. By increasing the size of the convolution kernel and the value of the expansion coefficient, the receptive field of the data is enlarged, and the longer convolution “memory” is formed. Figure 3a depicts its structure. More formally [28], for a one-dimensional sequence input x∈ ℝ
and a filter f:{0,…,k−1}→ℝ, the extended convolution operation F on sequence element s is defined as: 

(2)$$\;\;\;\;F\left(s\right)=\left(x\times df\right)\left(s\right)=\overset{k-1}{\underset{i=0}{\sum }}f\left(i\right)\cdot x_{s-d\cdot i}$$
where  d is the expansion coefficient, k is the filter size, and s−d·i denotes the past direction. As shown in Figure 3a, the expansion coefficient of layer I of the hidden layer is d=1. Layer II is d=2, which means that every two-time step is taken as an input. As the effective window size increases exponentially, larger receptive fields and lower network complexity can be obtained with fewer layers.

TCN introduces a residual network. The characteristic of a residual block is that it contains a branch that produces output by a series of transformations ℱ
and then adds its output to the input x of the block. Its core idea is to “connect” network layers separated by one or more layers to effectively solve the problem of gradient vanishing in complex models [33]:


(3)
o=Activation(x+ℱ(x))


In the T2Trans model, we exploit a generic residual module to replace the convolution layer. The remaining block of TCN is shown in Figure 3b. In the residual block of each layer, there are two layers of dilated causal convolution and the activation function ReLU [34]. ReLU can significantly speed up the training process and apply weight normalization to the convolution kernel. To account for the different input–output widths, we use an additional 1×1 convolution to ensure that the elements summing up receive tensors of the same shape, as shown in Figure 4. The initial inputs of the channel TCN layer are four tensors: $A_{n,\;500,3}$, $A_{n,\;500,3}$, $A_{n,\;500,3}$, and $A_{n,\;500,1}$. The four output tensors are all $A_{n,\;500,32}$, respectively. The four output tensors passing through the channel TCN layer are merged into $A_{n,\;500,128}$, which are fed into the fusion TCN layer.

3.Fusion TCN Layer. The fusion TCN layer model structure is similar to that of the channel TCN layer. The size of each convolution kernel is denoted by k=3
and d=1, 2. The output of $A_{n,\;500,128}$ from the channel TCN layer is fed into the fusion TCN layer, and the output tensor of the entire fusion TCN layer is $A_{n,\;500,32}$.

4.MLP Layer. In the MLP layer, feature representations from the fusion TCN layer are implicitly learned. The MLP layer consists of five fully connected networks. We use the dropout method [35] at the MLP layer to reduce the impact of overfitting problems [36] on the performance of T2Trans, and L2 regularizers further enhance the generalization capability of the model. We define the output of the *i*-th fully connected layer by the following formula:


(4)
$$\;o_{i}=\;\mathrm{Activation}\;\left(W\cdot x+b\right)$$


We define *W* as the weight of the hidden layer and *b* as the deviation.

In the MLP layer, the number of cells in all hidden layers is 128, 256, 512, 1024, and 8, respectively, and the activation functions are ReLU, ReLU, ReLU, ReLU, and Softmax, respectively. Set the dropout probability of hidden layer I, hidden layer II, hidden layer III, and hidden layer IV to 0.25, 0.5, 0.25, and 0.5, respectively. Set L2 regularizers of hidden layer II and hidden layer IV to 0.01 to avoid overfitting. Finally, hidden layer V uses Softmax as the activation function, and finally outputs the result $A_{n,\;8}$, as shown in Figure 5.

## 4. Experimental Evaluation

In this section, we conduct several experiments to evaluate the performance of T2Trans on the SHL dataset and HTC dataset. Firstly, the two datasets and the process of data processing in the experiment are introduced in detail. The performance of T2Trans is then compared to the baseline. Detailed settings for several baseline algorithms are listed below. Finally, the computational complexities of the algorithm and the comparison algorithm are evaluated.

### 4.1. Datasets

SHL Dataset: We chose the SHL dataset [7] to evaluate the performance of the model. First, it is transformed into an N×M
matrix, where N denotes the total number of samples collected and M denotes the total number of elements observed by different sensors. In particular, M=10 for the SHL dataset. After preprocessing the SHL dataset, we divided it into the training part (70%) and the test part (30%). We are now in a position to elaborate on the details of the SHL dataset. In 2017, three British volunteers spent seven months collecting their traffic data to form the SHL dataset. Eight modes of transportation were tagged during daily traffic transfers. Each sample contains ambient light, temperature, GPS, WiFi, and motion data. The samples were taken with Huawei Mate 9 smartphones placed in various locations, such as bags, and hands, strapped to the chest, or in pockets. In this paper, only lightweight sensor data (i.e., accelerometers, gyroscopes, magnetometers, and barometers) are leveraged to evaluate our proposed model to identify low-power transportation. In practical application scenarios, sensor data collected in the hand or in the pocket are more often employed, and, therefore, they were chosen for performance evaluation. We chose approximately 272 h of sensor data to train and test our model. The data were collected over four months by the same volunteer. All sensor data were sampled at 100 Hz. To take advantage of time dependence in our experiments, we rearranged the SHL data in chronological order.HTC Dataset: To evaluate the scalability of the T2Trans model, a baro-free large-scale dataset called the HTC dataset [8] is used. The HTC dataset has been collected since 2012. A total of 8311 h and 100 GB of data were collected by 150 volunteers using HTC smartphones. Each sample contains accelerometer, gyroscope, and magnetometer data. To maintain consistency among the three evaluation datasets, motorcycle and high-speed rail data from the HTC dataset were discarded. In contrast, different sensor data with timestamp differences of less than 0.1 s were defined. After preprocessing the HTC dataset, we divided it into the training part (70%) and the test part (30%).

### 4.2. Data Preprocessing

We use a fixed-length sliding window to split the raw data. We treat each window as a “sequence”, which is the initial input to T2Trans. We have chosen eight windows corresponding to each of the eight traffic modes. The raw data are then split into a series of fixed-length sequences. Figure 6 shows the data on the x-axis of the linear acceleration for the eight traffic modes in the SHL dataset and Figure 7 shows the data on the x-axis of the linear acceleration for the eight traffic modes in the HTC dataset.

The original sensor data contain various noises and errors, and the value range of different sensor data is different. To provide clean and normalized data for the T2Trans model, dirty data removal and data normalization operations are carried out before being fed into the model, which can effectively improve the training and prediction accuracy.

Dirty data removal: Considering the large size of the public dataset, we adopted a low-cost removal operation for the samples with incomplete sensor vector elements, that is, three-dimensional sensor data without one or two elements. For small datasets, interpolation may be more appropriate.Normalization: To deal with the large difference in the value range of heterogeneous sensor data, the z-fraction normalization operation [37] is applied to each element of sensor vector data, and the formula is as follows:

(5)$$x^{\prime }=\frac{x-\mu }{\sigma }$$
where  μ is the average value of each element of sensor data and σ is the standard deviation of each element of sensor data.

### 4.3. Baseline

To evaluate the performance of our proposed T2Trans, several algorithms were utilized as baselines and benchmarks for comparative experiments, including classical machine learning algorithms (i.e., DT and RF), and deep learning algorithms (i.e., CNN and CNN + LSTM). Among these baselines, CNN is a part of our proposed model. DT and RF are implemented using the MATLAB machine learning toolbox, and all classifiers use default parameters set in library functions unless explicitly mentioned. In DT, the researcher set the parameter “Min_leaf_Size = 1000”. In RF, the researcher set the number of trees to 20 and set the parameter “min_leaf_size = 1000” in each tree. Table 1 lists the detailed parameters of all baselines.

DT: Decision Tree is a commonly used classification and regression method [38].RF: Random Forest is a relatively new machine learning model [39].XGBOOST: XGBOOST is an optimization to Boosting, which integrates weak classifiers into a strong one. The XGBOOST algorithm generates a new tree to fit the residual of the previous tree through continuous iteration. With the increase in iteration time, the accuracy keeps improving [40].CNN: Convolutional networks learn what is actually a local pattern by convolutional operations on the local area. In this way, increasingly complex and abstract visual concepts can be learned through multiple convolutions.MLP: Adopt a set of multilayer perceptions. Each perceptron is trained with data from different specific smartphone locations, including small datasets of hand-held phones. An iterative reweighting scheme is proposed to combine classifiers, which considers their consistency with specialized hand classifiers.LR + MLP [41]: The application of LR and MLP neural network models to enhance the predictive power of the model. Logistic regression answers the “whether” question. We add a Softmax to linear regression for multiple classifications and use a cross-entropy loss function. LR is a linear model and MLP is a nonlinear model; MLP fits are more complex.Bi-LSTM: The recursive neural network method is used. The two-way LSTM architecture was proposed to address this challenge. The model was trained based on rotation and translational invariance to ignore the orientation and position information of the smartphone [19].CNN + LSTM: It combines CNN and LSTM [42]. In this method, CNN allows the learning of feature representations suitable for recognition, and these feature representations are robust for transportation mode detection. LSTM unit is applied to the output of CNN, which plays a role in structural dimension reduction on feature vectors.

### 4.4. Metrics

Accuracy, precision, recall, and F1-score were used to evaluate T2Trans, and F1-score was defined as follows:


(6)
$$F1=2\times \frac{\;\mathrm{Precision}\;\times \;\mathrm{Recall}\;}{\;\mathrm{Precision}+\mathrm{Recall}\;}$$


### 4.5. Experimental Settings

We exploit the Keras deep learning framework to train T2Trans with the preprocessed SHL dataset. We adopt the Adam [43] optimizer and cross-entropy loss function for optimization with learning rate = 0.001, beta1 = 0.9, and beta2 = 0.999. The epoch parameter is set to 150 and the batch size is set to 32. We conducted model training on PC with GPU support. Table 2 shows the detailed configuration of the PC.

### 4.6. Experimental Results of Different Baselines on SHL Dataset and HTC Dataset

As shown in Table 3 and Table 4, we can observe that: 3.The results of CNN, CNN + LSTM, and T2Trans are generally superior to traditional machine learning methods because CNN + LSTM and T2Trans make full use of the advantages of CNN in feature extraction and give full play to the advantages of deep learning. Classic machine learning hand-extracted features may not do a good job of distinguishing between train and subway patterns. The precisions of MLP and LR + MLP are also above 70%, but the performance is similar to that of machine learning algorithms. High-level features or time dependencies may not be learned using these baselines.4.CNN + LSTM is better than other methods, indicating that CNN can learn appropriate feature representations for identification, and these feature representations are robust to transportation mode detection. LSTM unit is used on CNN output, which plays a role in structural dimension reduction on the feature vector, thus significantly improving the performance of transportation mode detection.5.Our proposed T2Trans was significantly superior to other baselines. The F1 score of the model on the SHL dataset is 86.42% and on the HTC dataset is 88.37%. The accuracy of the algorithms based on DT, RF, XGBOOST, MLP, LR + MLP, Bi-LSTM, and CNN + LSTM are all above 70%. Nevertheless, these baselines do not distinguish well between train and subway modes with high precision. T2Trans not only uses the time convolution layer to construct the entire network, but also includes convolution and pooling operations, and the construction of remaining cells in the complete connection layer not only speeds up the training and prediction process but also improves the overall performance of transportation mode detection. As illustrated in Table 4, classical machine learning algorithms can accurately recognize most transportation modes, i.e., still, walk, run, bike, car, and bus. However, the accuracy for train and subway is lower. Instead, a reasonable representation of all eight modes of transportation was obtained using T2Trans and better accuracy was achieved in all the above baselines.

The SHL dataset and the HTC dataset were each divided into two datasets, namely the training dataset (70%) and the testing dataset (30%). Table 5 shows the accuracy performance of the models built on the two datasets. When using the SHL dataset, the average accuracy of the T2Trans model on the training dataset was 95.56% and on the test dataset was 86.37%. When using the HTC dataset, the average accuracy of the T2Trans model was 96.66% on the training dataset and 89.13% on the test dataset. From the above results, it can be seen that when using the T2Trans algorithm, the average accuracy on the HTC training and test datasets is approximately 1.1% and 2.8% higher than that on the training and test datasets of the SHL dataset, respectively. Meanwhile, the accuracy of the test dataset is usually lower than that of the training set, but the average accuracy of the test dataset is above 86%, and the T2Trans algorithm has good performance.

In addition, Figure 8 illustrates the confusion matrix for the model on the SHL dataset. Figure 9 illustrates the confusion matrix for the model on the HTC dataset. Figure 10 and Figure 11 show the precision, recall, and F1-score of T2Trans on the SHL dataset and HTC dataset, respectively. The precision and recall metrics are commonly used for evaluating multiple classification problems. F1 score is a harmonic average of precision and recall.

To better demonstrate the performance of T2Trans, we have cited several papers on transportation mode detection that have worked well in recent years. Table 6 and Table 7 compare the accuracy of the T2Trans algorithm and these algorithms on the SHL dataset and the HTC dataset, respectively. DL involves four classifiers: Bi-LSTM [19], CNN + LSTM [22], convolutional neural network [23] and LR + MLP [41]. It can be seen from the performance on both datasets that although ML has a lower upper bound than DL, the lower bound is higher as ML has better robustness as it utilizes hand-crafted features that can cope with user and location variations. In contrast, features learned by DL do not always guarantee good generalization. ML involves three classifiers: the researchers mainly used the RF [9], MLP algorithms [20], and XGBOOST [40]. Of these algorithms for transportation mode detection, CNN showed the highest average accuracy, followed by CNN + LSTM. On the SHL dataset and the HTC dataset, our T2Trans model produced results that improved accuracy by 2.67% and 2.4% over the best CNN algorithm of these algorithms.

### 4.7. Hyperparameter Fine-Tuning

Hyperparameter fine-tuning is important to improve the performance of T2Trans. T2Trans has a total of eight convolutional layers. The different hyperparameter configurations are grouped into the eight groups a-h. F and K represent filters and kernel size of all eight convolutional layers, respectively, on the SHL dataset. For Group a, F = [64, 64, 32, 32, 32, 32, 32, 32] and K = [3, 3, 3, 3, 3, 3, 3, 3]. For Group b, F = [32, 32, 32, 32, 32, 32, 32, 32] and K = [2, 2, 3, 3, 3, 3, 3, 3]. For Group c, F = [32, 32, 32, 32, 32, 32, 32, 32] and K = [2, 2, 2, 2, 3, 3, 3, 3]. For Group d, F = [32, 32, 32, 32, 64, 64, 32, 32] and K = [3, 3, 3, 3, 3, 3, 3, 3]. For Group e, F = [32, 32, 32, 32, 32, 32, 64, 64] and K = [3, 3, 3, 3, 3, 3, 2, 2]. For Group f, F = [32, 32, 32, 32, 128, 128, 32, 32] and K = [3, 3, 3, 3, 3, 3, 3, 3]. For Group g, F = [64, 64, 32, 32, 64, 64, 64, 64] and K = [3, 3, 3, 3, 2, 2, 3, 3]. For Group h, F = [32, 32, 32, 32, 64, 64, 64, 64] and K = [2, 2, 2, 2, 3, 3, 3, 3]. 

Figure 12 depicts the F1 scores for different hyperparameter configurations, and Group “e” obtains the best result. The train and subway modes are the most challenging of the eight modes of transportation. This is caused by similar transportation mode detection, namely smooth running on railway tracks.

### 4.8. Impact of Different Sensor Components

To explore the scalability of T2Trans, we added different sensor variables to achieve the effect of research identification. As shown in Figure 13, for the SHL dataset, the average transportation mode detection accuracies of LA, LAG, LAGM, and LAGMP were 76.68%, 79.23%, 81.93%, and 86.42%, respectively. By adding sensors, the accuracy of models was improved by about 2.6 to 4.5%. As shown in Figure 14, for the HTC dataset, the average transportation mode detection accuracies of LA, LAG, LAGM, and LAGMP are 82.29%, 83.21%, 86.07%, and 89.13%, respectively. By adding sensors, the accuracy of models was improved by about 0.9 to 3.0%. As more sensor variables are used, T2Trans may learn more about features, especially the barometer sensor for trains and the subway.

### 4.9. Calculation Complexity

The computer we used is equipped with an Intel(R) Xeon(R) CPU @ 2.30ghz and a Tesla P100-PCI-16GB GPU. The code is written in Python 3.7. Table 8 compares the results of various classifiers in terms of processing time and parameter size. The training time for MLP, CNN, CNN + LSTM, and T2Trans is calculated for one hundred and forty epochs.

As shown in Table 8, although the training time of the T2Trans algorithm is longer than that of DT, RF, and XGBOOST classical machine learning, because T2Trans uses deep learning, it needs a lot of time to learn data features. However, T2Trans has significantly improved performance by automatically learning data features. Although the training parameters of the T2Trans algorithm are larger than those of LSTM and other standard cyclic networks, the training time is smaller than LSTM. The reason why the T2Trans algorithm is faster than LSTM is that the forward transmission process of input information of all time steps of TCN is carried out simultaneously and can be completed in parallel. However, the RNN network needs to wait for the end of the forward transmission of the previous time step before the forward transmission of the next time step. T2Trans is more accurate and concise than LSTM and other standard circulation networks because T2Trans introduces dilated convolution and residual connections.

### 4.10. The Future Work

Overall, T2Trans is a variant of a convolutional neural network. Although the use of extended convolution can expand the perceptual field, it is still limited and inferior to the Transformer, which can capture relevant information of arbitrary length. In our future work, We will draw on the Transformer’s ability to grab relevant information of arbitrary length to improve T2Trans and make T2Trans shine in other areas as well.

## 5. Conclusions

In this paper, we propose a novel deep learning model for transportation mode detection. A TCN is exploited to automatically learn the features of multiple lightweight sensors integrated into smartphones to detect transportation modes, which can achieve an energy-saving effect. T2Trans also exhibits very good recognition performance, as it achieves an average precision of 86.57% on the SHL dataset and 89.25% on the HTC dataset. This is an improvement of 3.01% and 6.97% over the performance of the best baseline, respectively. By learning the features from different sensors in different channel TCNs, more heterogeneous sensors can be supported. Moreover, TCN can be processed in parallel on a large scale, with more flexibility in the length of historical information and less memory for training, especially for long sequences. As a result, T2Trans will be faster and more practical for real-world transportation mode detection. As part of our future work, we plan to port T2Trans to mobile phones to evaluate the generalization capabilities of our model in real applications.

## Figures and Tables

**Figure 1 sensors-22-06712-f001:**
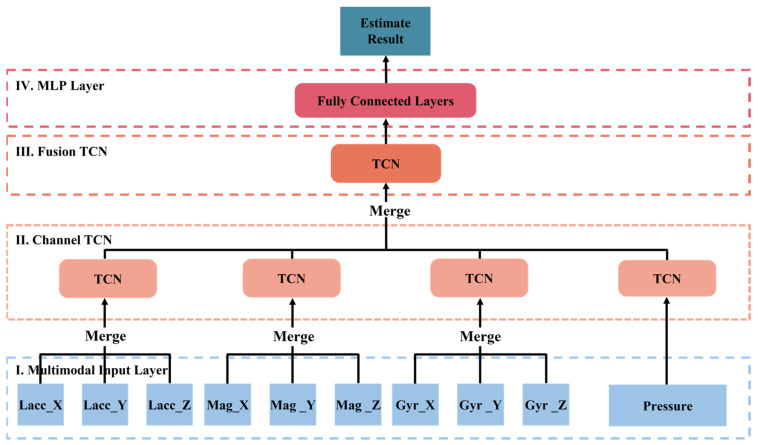
The schema of the overall overview of our proposed T2Trans model. Preprocessed sensor data are fed into the model from the multimodal input layer, which are fed into ten channel TCN layers, respectively. The fusion TCN layer model structure is similar to that of the channel TCN layer. In the MLP layer, feature representations from the fusion TCN layer are implicitly learned.

**Figure 2 sensors-22-06712-f002:**
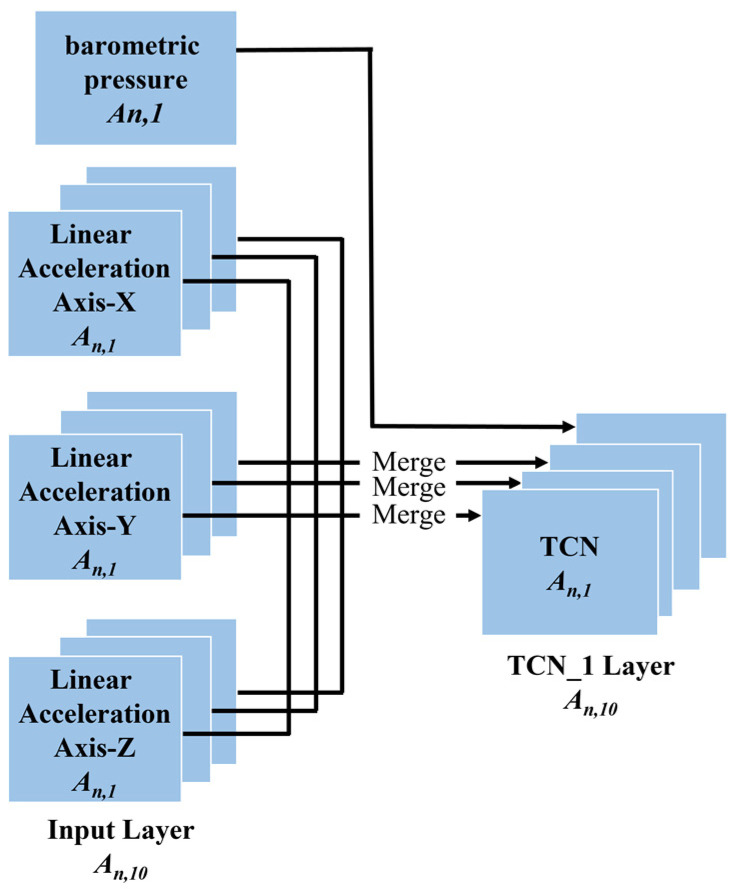
The overview of the proposed multimodal input layer. A_(n,d,10) is converted into ten tensors denoted by, which are fed into ten channel TCN layers, respectively.

**Figure 3 sensors-22-06712-f003:**
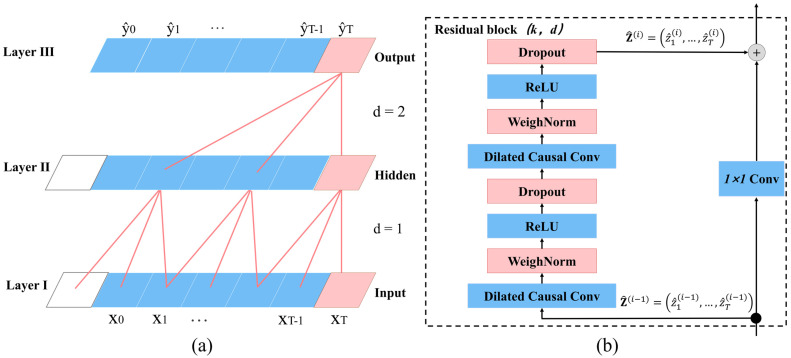
Architectural elements in a TCN. (**a**) A dilated causal convolution with dilation factors d=1, 2. The perceptual field is able to cover all values of the input sequence. (**b**) When the remaining inputs and outputs have different sizes, a 1 × 1 convolution is added. Inputs and outputs have different sizes.

**Figure 4 sensors-22-06712-f004:**
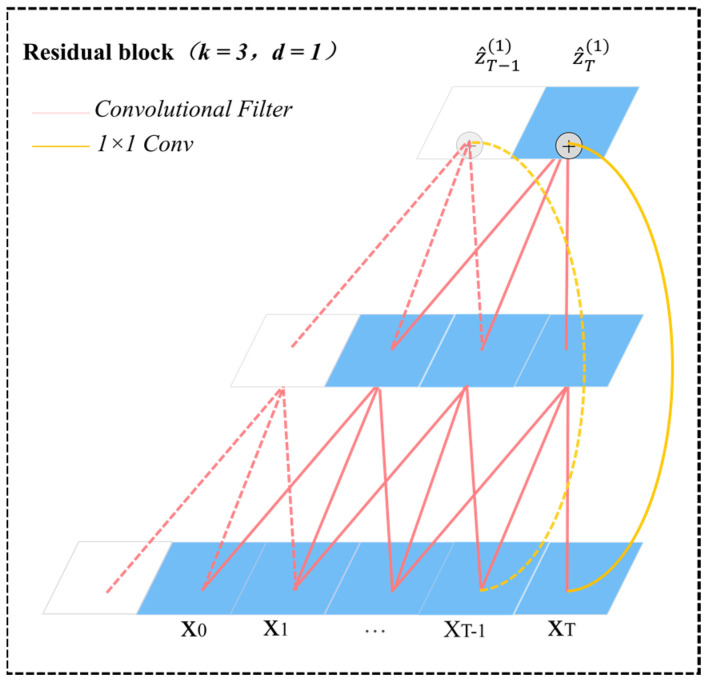
An example of residual connection in a TCN. An additional 1 × 1 convolution is added to ensure that the elements summing up receive tensors of the same shape. The red lines are filters in the residual function and the yellow lines are identity mappings.

**Figure 5 sensors-22-06712-f005:**
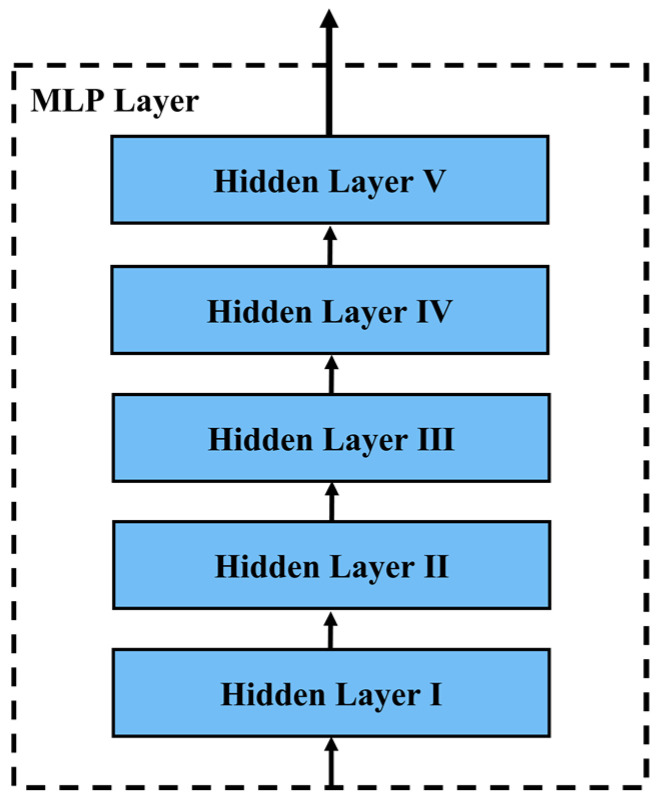
The architecture of the MLP layer. The MLP layer consists of five fully connected networks. Hidden layer V uses Softmax as the activation function and finally outputs the result.

**Figure 6 sensors-22-06712-f006:**
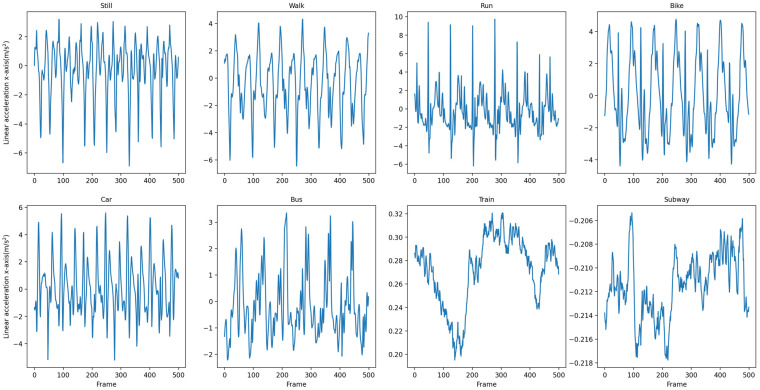
Data on the x-axis of the linear acceleration in 8 traffic modes in the SHL dataset.

**Figure 7 sensors-22-06712-f007:**
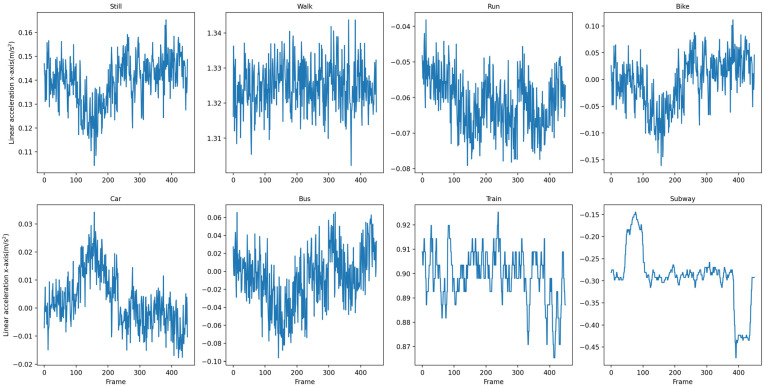
Data on the x-axis of the linear acceleration in 8 traffic modes in the HTC dataset.

**Figure 8 sensors-22-06712-f008:**
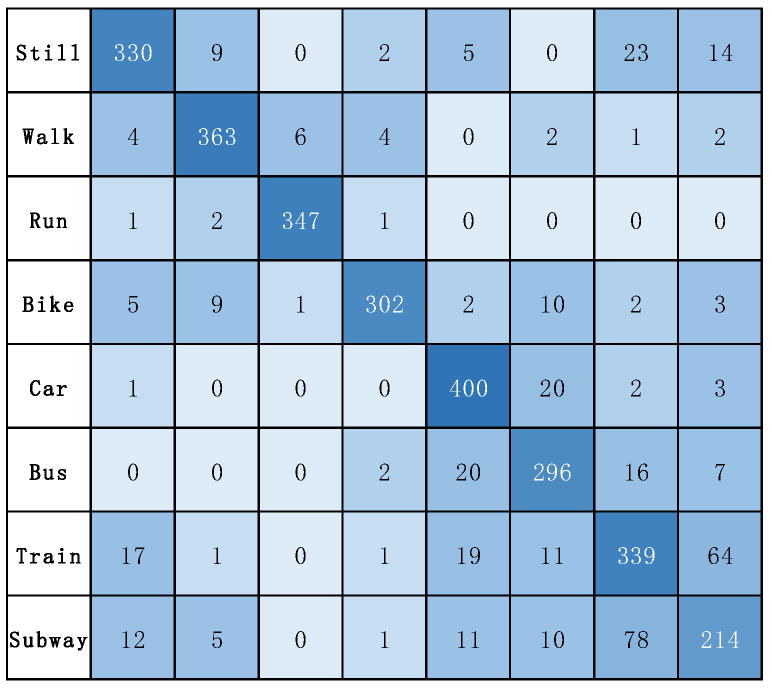
The confusion matrix of T2Trans on the SHL dataset.

**Figure 9 sensors-22-06712-f009:**
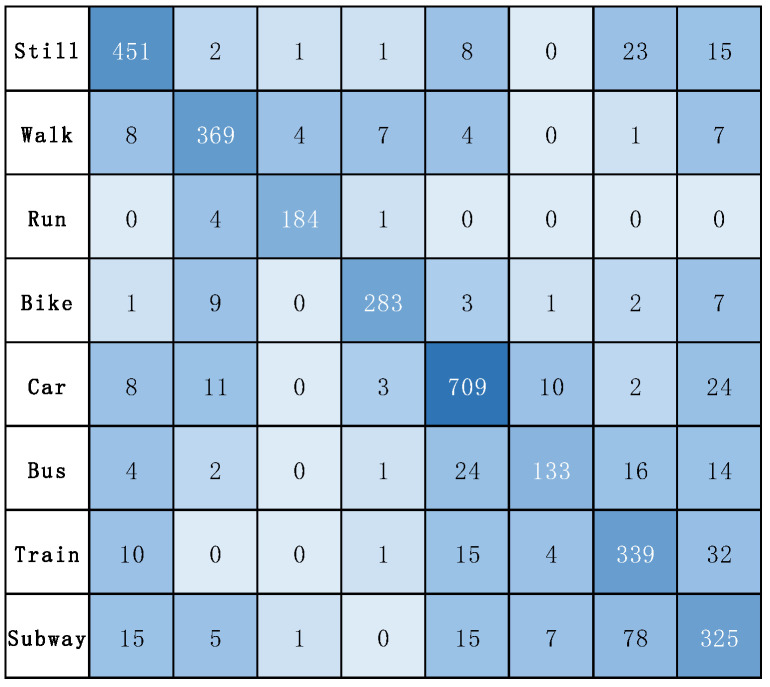
The confusion matrix of T2Trans on the HTC dataset.

**Figure 10 sensors-22-06712-f010:**
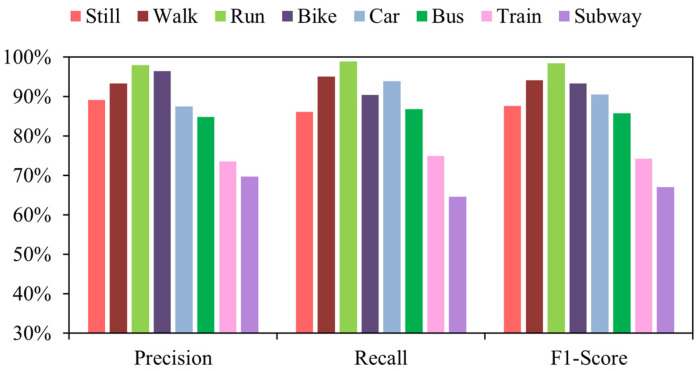
Precision, recall, and F1-scores of the eight classifications of the T2Trans model on the SHL dataset.

**Figure 11 sensors-22-06712-f011:**
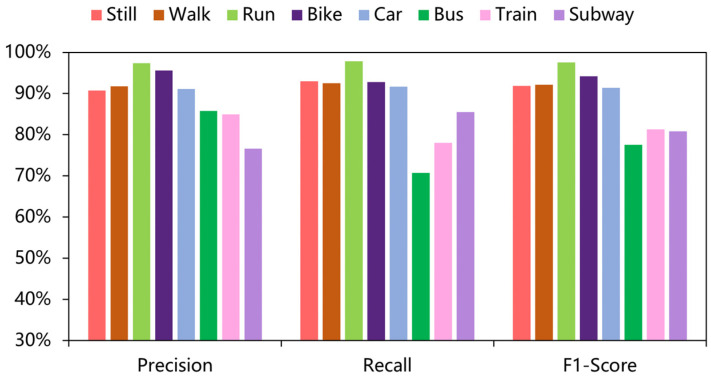
Precision, recall, and F1-scores of the eight classifications of the T2Trans model on the HTC dataset.

**Figure 12 sensors-22-06712-f012:**
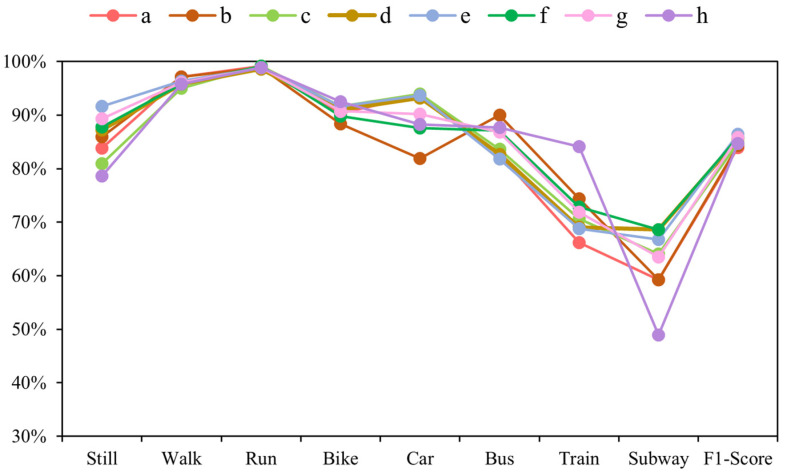
F1-Scores obtained on the SHL dataset using eight different sets of hyperparameters, a-h, for transportation mode detection.

**Figure 13 sensors-22-06712-f013:**
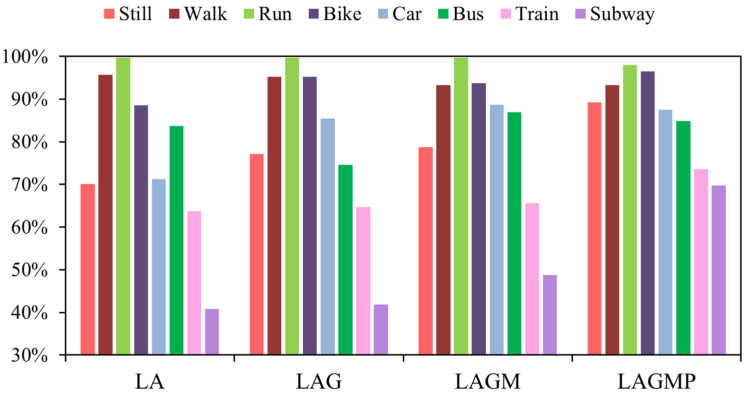
The precisions of T2Trans using different sensors on the SHL dataset. Note that LA is short for linear accelerometer, LAG is short for linear accelerometer + gyroscope, LAGM is short for linear accelerometer + gyroscope + magnetometer, and LAGMP is short for linear accelerometer + gyroscope + magnetometer + barometric pressure.

**Figure 14 sensors-22-06712-f014:**
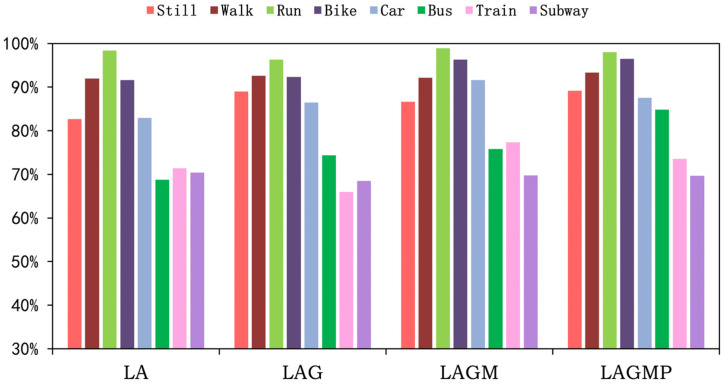
The precisions of T2Trans using different sensors on the HTC dataset. Note that LA is short for linear accelerometer, LAG is short for linear accelerometer + gyroscope, LAGM is short for linear accelerometer + gyroscope + magnetometer, and LAGMP is short for linear accelerometer + gyroscope + magnetometer + barometric pressure.

**Table 1 sensors-22-06712-t001:** The detailed parameters of the baselines.

Name	Architecture
DT	min_leaf_size = 1000
RF	TreeBagger: NumTrees = 20, minleafsize = 1000
XGBOOST	n_estimators = 900, max_depth = 7, min_child_weight = 1
CNN	C(32)-C(32)-C(64)
MLP	FC(128)-FC(256)-FC(512)-FC(1024)-Softmax
Bi-LSTM	LSTM(128)-LSTM(128)-FC-Softmax
CNN + LSTM	EACH ELEMENT[C(64)-C(128)]-C(32)-LSTM(128)-DNN(128)-DNN(256)-DNN(512)-DNN(1024)-Softmax

Note: FC is a fully connected layer; C is a convolutional 1D layer; P is a MaxPooling 1D layer.

**Table 2 sensors-22-06712-t002:** The configuration of PC.

Name	Detail
CPU	Intel(R) Xeon(R) CPU @ 2.30 GHz
Memory	16GB
GPU	Tesla P100-PCIE-16GB
Operating System	Ubuntu 18.04.5 LTS
Python Environment	3.7.13
Development Framework	Keras

**Table 3 sensors-22-06712-t003:** The precisions of different transportation mode detection methods using different algorithms on the SHL dataset.

	DT	RF	XGBOOST	CNN	MLP	CNN + LSTM	T2Trans
Still	72.87%	74.75%	75.60%	90.97%	75.45%	82.14%	89.18%
Walk	76.01%	86.77%	89.98%	95.97%	87.43%	92.82%	93.31%
Run	87.58%	97.68%	99.34%	98.02%	96.30%	98.50%	98.02%
Bike	72.32%	83.56%	85.10%	95.05%	67.37%	93.81%	96.49%
Car	68.93%	65.42%	72.76%	85.81%	65.49%	79.67%	87.53%
Bus	60.30%	57.73%	61.11%	77.27%	62.17%	68.83%	84.81%
Train	66.57%	59.78%	64.46%	62.21%	56.64%	68.78%	73.54%
Subway	61.97%	54.51%	52.63%	63.16%	62.24%	60.49%	69.70%

**Table 4 sensors-22-06712-t004:** The precisions of different transportation mode detection methods using different algorithms on the HTC dataset.

	DT	RF	XGBOOST	CNN	MLP	CNN + LSTM	T2Trans
Still	76.74%	93.22%	81.91%	91.76%	82.47%	86.31%	90.74%
Walk	54.12%	68.04%	79.41%	89.66%	62.54%	81.03%	91.79%
Run	77.98%	94.83%	93.62%	97.34%	86.22%	94.76%	97.35%
Bike	52.06%	75.42%	79.74%	90.79%	65.54%	83.89%	95.61%
Car	66.23%	74.50%	69.36%	84.85%	80.94%	82.99%	91.13%
Bus	36.07%	84.48%	74.71%	62.50%	73.01%	71.22%	85.81%
Train	58.38%	78.53%	74.50%	69.44%	69.88%	76.60%	84.94%
Subway	50.42%	75.22%	68.36%	71.88%	64.35%	78.99%	76.65%

**Table 5 sensors-22-06712-t005:** Accuracy performances for training datasets and test datasets.

	SHL DATASET		HTC DATASET	
	Training	Test	Training	Test
Still	97.62%	86.16%	95.87%	92.99%
Walk	98.72%	95.03%	97.59%	92.48%
Run	99.93%	98.86%	99.56%	97.87%
Bike	99.26%	90.42%	98.60%	92.79%
Car	99.48%	93.90%	97.73%	91.72%
Bus	99.53%	86.80%	84.67%	70.74%
Train	95.10%	75.00%	91.99%	78.01%
Subway	83.65%	64.65%	98.50%	85.53%

**Table 6 sensors-22-06712-t006:** Performance comparison between the algorithms of the best teams and the T2Trans algorithm on the SHL dataset for the third SHL recognition challenge.

	RF	XGBOOST	CNN	MLP	LR + MLP	Bi-LSTM	CNN + LSTM	T2Trans
Still	75.95%	85.40%	81.46%	84.74%	86.18%	72.45%	80.16%	86.16%
Walk	89.96%	84.65%	96.34%	60.15%	79.70%	93.62%	94.76%	95.03%
Run	89.56%	89.43%	98.29%	87.77%	85.64%	98.91%	98.29%	98.86%
Bike	83.63%	64.90%	94.31%	57.38%	80.33%	91.40%	88.92%	90.42%
Car	58.18%	90.09%	90.38%	83.31%	76.33%	76.82%	81.92%	93.90%
Bus	60.59%	38.28%	76.83%	59.57%	48.93%	60.67%	69.21%	86.80%
Train	58.60%	69.95%	66.59%	80.85%	60.99%	52.49%	54.20%	75.00%
Subway	63.37%	68.83%	65.26%	46.84%	65.53%	55.34%	57.10%	64.65%

**Table 7 sensors-22-06712-t007:** Performance comparison between the algorithms of the best teams and the T2Trans algorithm on the HTC dataset for the third SHL recognition challenge.

	RF	XGBOOST	CNN	MLP	LR + MLP	Bi-LSTM	CNN + LSTM	T2Trans
Still	93.90%	89.14%	89.69%	87.22%	84.33%	84.94%	89.07%	92.99%
Walk	68.27%	75.44%	91.23%	67.42%	75.69%	78.47%	86.72%	92.48%
Run	94.74%	92.05%	97.87%	92.55%	92.55%	88.88%	94.15%	97.87%
Bike	73.02%	69.77%	89.84%	67.21%	82.95%	82.13%	87.87%	92.79%
Car	75.04%	81.56%	88.23%	81.50%	78.00%	75.31%	83.83%	91.72%
Bus	92.59%	52.68%	70.74%	62.23%	42.02%	62.50%	58.51%	70.74%
Train	78.88%	73.99%	76.95%	77.31%	47.16%	67.01%	75.18%	78.01%
Subway	73.50%	71.88%	78.42%	61.05%	67.37%	56.78%	71.05%	85.53%

**Table 8 sensors-22-06712-t008:** The training and predicting time of different algorithms.

Algorithm	Platform Type	Training Time	Parameter Size
DT	GPU	90 s	-
RF	GPU	90 s	-
XGBOOST	GPU	6653.52 s	-
MLP	GPU	140.44 s	1,338,248
CNN	GPU	1200 s	70,348
CNN + LSTM	GPU	64,080 s	125,568
T2Trans	GPU	786.09 s	796,880

## Data Availability

The SHL dataset for this article is publicly available on the website http://www.shl-dataset.org/dataset (accessed on 1 May 2022) and the HTC dataset for this article is required to see reference [8].

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
