# Peer review of "Transportation Mode Detection Using Temporal Convolutional Networks Based on Sensors Integrated into Smartphones"

_sensors, 2022, doi:10.3390/s22176712_

Round 1

Reviewer 1 Report

This paper proposed deep learning algorithms to identify the transportation mode of road users carrying a smartphone. The data used are from the IMU sensor, including accelerometers, gyroscopes, magnetometers, and barometers, in contrast to the previous studies using GPS traces for cellular vehicle probes. The model prediction results outperform the previous models.

However, it is unclear the application of the presented study. How can the proposed methodology be applied to detect the modes of travelers in the road system? IMU data is not to be collected by the mobile service provider. Or if the algorithm is to be installed on the smartphone? The smartphone user knows the transportation mode they are using, and it seems there is no need to detect the mode automatically. Please clarify the potential application of this methodology.

Author Response

We have uploaded our response letter as a PDF file, please refer to it. Thank you.

Reviewer 2 Report

Refer to the attached file, Review_R1_Trans_mode, please.

Author Response

(The authors gave the same response as above.)

Reviewer 3 Report

The paper with title, ‘Transportation Mode Detection Using Temporal Convolutional  Networks Based on Sensors Integrated in Smartphones’ transportation mode detection algorithm has been proposed, which is namely as T2Trans. The model is based on temporal convolutional networks, such as TCN by using myriad lightweight sensors being integrated with phone.  Furthermore, results of F1-score of 86.42% on the real world SHL dataset and 88.37% on the HTC dataset are achieved. The carried out work is interesting and quite comprehensive. However, before acceptance following comments must be incorporated.

1.       The abstract and Conclusion could be improved by adding more quantitative data (results).

2.       More recent works should be added in the citations.

3.       Very brief captions are given to all figures, authors should add little details in each figure.

4.       There are various grammatical mistakes, which should be corrected in the revised manuscript.

5.       The variable in equation (1) need explanation.

6.       Authors should add a comparison table which shows the standing of their achieved results in comparison to the recently reported studies.

7.       It would be better, if authors add a paragraph before conclusion about the future work (as this algorithm still needs improved to be at mature level).

Author Response

(The authors gave the same response as above.)

Round 2

Reviewer 2 Report

Refer to the attached file, please.

Author Response

(The authors gave the same response as above.)

Reviewer 3 Report

All of my comments are well addressed. So I recommend its acceptance.

Author Response

Thanks for your insightful comments.